# A Cancer-Specific Anti-Podocalyxin Monoclonal Antibody (humPcMab-60) Demonstrated Antitumor Efficacy in Pancreatic and Colorectal Cancer Xenograft Models

**DOI:** 10.3390/antib14030067

**Published:** 2025-08-11

**Authors:** Hiroyuki Suzuki, Tomokazu Ohishi, Takuro Nakamura, Miyuki Yanaka, Saori Handa, Tomohiro Tanaka, Mika K. Kaneko, Yukinari Kato

**Affiliations:** 1Department of Antibody Drug Development, Tohoku University Graduate School of Medicine, 2-1 Seiryo-Machi, Aoba-ku, Sendai 980-8575, Japan; takuro.nakamura.a2@tohoku.ac.jp (T.N.); miyuki.yanaka.c5@tohoku.ac.jp (M.Y.); saori.handa.d3@tohoku.ac.jp (S.H.); tomohiro.tanaka.b5@tohoku.ac.jp (T.T.); mika.kaneko.d4@tohoku.ac.jp (M.K.K.); 2Institute of Microbial Chemistry (BIKAKEN), Laboratory of Oncology, Microbial Chemistry Research Foundation, 3-14-23 Kamiosaki, Shinagawa-ku, Tokyo 141-0021, Japan; ohishit@bikaken.or.jp

**Keywords:** podocalyxin, cancer-specific monoclonal antibody, antibody-dependent cellular cytotoxicity, complement-dependent cellular cytotoxicity, human cancer xenograft

## Abstract

**Background**: Podocalyxin (PODXL) has been identified as a promising therapeutic target and a potential diagnostic biomarker in various tumors. Despite the therapeutic potential of anti-PODXL monoclonal antibodies (mAbs), their further development has been limited by concerns regarding potential on-target off-tumor toxicities. To minimize adverse effects on normal tissues, developing a cancer-specific mAb (CasMab) against PODXL is essential. **Methods**: Our group established a cancer-specific anti-PODXL mAb, PcMab-60 (IgM, κ), through the screening of over one hundred hybridoma clones. In this study, PcMab-60 was engineered into a humanized IgG_1_-type mAb (humPcMab-60), and its antitumor activity was examined using mouse xenograft models of pancreatic ductal adenocarcinoma (PDAC) and colorectal cancer. **Results**: HumPcMab-60 retains cancer-specific reactivity; humPcMab-60 reacted to PDAC cell lines (PK-45H and MIA PaCa-2) and the colorectal cancer cell line (Caco-2), but not to a normal lymphatic endothelial cell line in flow cytometry. Furthermore, humPcMab-60 exerted antibody-dependent cellular cytotoxicity and complement-dependent cytotoxicity against PODXL-expressing cell lines and showed antitumor effects against the tumor xenografts. **Conclusions**: A humanized anti-PODXL CasMab, humPcMab-60, could be a promising mAb-based tumor therapy.

## 1. Introduction

Podocalyxin (PODXL) is a transmembrane glycoprotein that belongs to the CD34 family [1]. The core protein of PODXL has a molecular weight of approximately 53,000 and undergoes extensive post-translational modifications such as *N*- and *O*-linked glycosylation, resulting in a mature glycoprotein with an apparent molecular weight ranging from 150,000 to 200,000 [2]. Under physiological conditions, PODXL is expressed in early hematopoietic progenitors [3], kidney podocytes [4], and vascular/lymphatic endothelial cells in adults [5]. PODXL has been implicated in the maintenance of homeostasis, and PODXL-deficient mice exhibit embryonic lethality [6]. Notably, aberrant overexpression of PODXL has been documented across a wide spectrum of human malignancies, including pancreatic ductal adenocarcinoma (PDAC) [7], renal cell carcinoma [8], colorectal cancer [9], breast cancer [10], and oral squamous cell carcinoma [11]. Elevated expression of PODXL is significantly correlated with poor disease-free survival, cancer-specific survival, and overall survival in colorectal cancer, PDAC, renal cell carcinoma, urothelial bladder cancer, and glioblastoma multiforme [12].

PODXL expression is markedly upregulated during the process of epithelial–mesenchymal transition [13]. PODXL plays a pivotal role in facilitating the extravasation of mesenchymal-type PDAC cells [14]. Mechanistically, PODXL promotes extravasation through interaction with a cytoskeletal linker protein, ezrin. The PODXL and ezrin interaction has been reported to stimulate intracellular signal transductions including mitogen-activated protein kinase, phosphatidylinositol-3 kinase, RhoA, Rac1, and Cdc42 pathways to promote motility [15]. Morphologically, the interaction supports the transition of tumor cells from a rounded and non-polarized phenotype to an invasive and extravasation-competent state [14]. These findings imply that PODXL is a key mediator of tumor cell extravasation during metastatic cascade. However, the involvement of an endothelial PODXL ligand, such as E-selectin [16], in this process remains to be elucidated. Studies involving both gain- and loss-of-function approaches have demonstrated that PODXL plays a critical role in tumor progression by enhancing cellular migration, invasiveness, stem cell-like properties, and metastasis across diverse cancer types [17]. Consequently, PODXL has emerged as a promising candidate for targeted tumor immunotherapy [15,18].

Our group previously generated an anti-PODXL monoclonal antibody (mAb), PcMab-47 (mouse IgG_1_, κ), for flow cytometry and immunohistochemistry [19]. To enhance its effector functions, PcMab-47 was engineered into a mouse IgG_2a_ isotype, designated 47-mG_2a_, thereby conferring antibody-dependent cellular cytotoxicity (ADCC) activity. Furthermore, to potentiate ADCC, a core fucose-deficient variant of 47-mG_2a_, termed 47-mG_2a_-f, was developed. Both 47-mG_2a_ and 47-mG_2a_-f demonstrated significant antitumor activity in mouse xenograft models of oral squamous cell carcinomas [20]. Other preclinical studies of anti-PODXL mAbs have also shown promising antitumor efficacy. A clone PODO83/PODOC, core protein-binding anti-PODXL mAb, suppressed breast cancer MDA-MB-231 xenograft growth and blocked lung metastasis [18].

Despite the therapeutic potential of anti-PODXL mAbs, their further development has been limited by concerns regarding potential on-target off-tumor toxicities, particularly to normal kidney podocytes [4] and vascular and lymphatic endothelial cells [5]. To minimize adverse effects on normal tissues, the development of cancer-specific mAbs (CasMabs) against PODXL is essential.

To address the challenge of tumor-specific targeting, our group developed CasMabs against various antigens to identify cancer-specific epitopes and elucidate their recognition mechanisms. In the case of human epidermal growth factor receptor 2 (HER2), over 300 anti-HER2 mAb clones were generated by immunizing mice with cancer cell-expressed HER2 and screened for selective reactivity via flow cytometry. Among them, H_2_CasMab-2 (H_2_Mab-250) selectively recognized HER2 on breast cancer cells but not on normal epithelial cells from the mammary gland [21]. Epitope mapping revealed that Trp614 in extracellular domain 4 of HER2 is essential for its binding [21]. A humanized H_2_CasMab-2 exhibited potent ADCC, complement-dependent cytotoxicity (CDC), and antitumor activity in breast cancer xenograft models [22]. Additionally, a single-chain variable fragment derived from H_2_CasMab-2 was applied to CAR-T cell therapy, demonstrating cancer-specific recognition and cytotoxicity [23]. A phase I clinical trial targeting HER2-positive advanced solid tumors is currently ongoing in the United States (NCT06241456). These findings highlight the importance of CasMab selection and epitope specificity in the development of effective therapeutic antibodies and related immunotherapies.

Using the same strategy, our group established a cancer-specific anti-PODXL mAb, PcMab-60 (IgM, κ), through screening over one hundred hybridoma clones. In flow cytometry, PcMab-60 reacted with the PODXL-overexpressed LN229 and pancreatic cancer MIA PaCa-2. In contrast, PcMab-60 did not recognize normal endothelial cells [24]. The epitope of PcMab-60 was demonstrated to be a peptide sequence in PODXL [25].

In this study, PcMab-60 was engineered into a humanized IgG_1_-type mAb (humPcMab-60), and its antitumor activity against PDAC and colorectal cancer xenografts was examined.

## 2. Materials and Methods

### 2.1. Cell Lines

Chinese hamster ovary (CHO)-K1, human colorectal carcinoma cell line (Caco-2), PDAC cell lines (PK-45H and MIA PaCa-2), and human lymphatic endothelial cell line (HDMVEC/TERT164-B) were obtained as described previously [26]. These cell lines and CHO/PODXL [19] were cultured as described previously [26].

### 2.2. Recombinant Antibody Production

Mouse anti-PODXL mAbs, such as PcMab-47 (IgG_1_, kappa) [19] and PcMab-60 (IgM, kappa) [25] were generated as previously described. To construct humanized versions (humPcMab-47 and humPcMab-60), the complementarity-determining regions (CDRs) of the variable heavy (V_H_) chains of PcMab-47 or PcMab-60 were grafted onto human IgG framework sequences (accession number: KF698734) and cloned into the pCAG-Neo expression vector along with the constant region (C_H_) of human IgG_1_. Similarly, the CDRs of the variable light (V_L_) chains, the human IgG framework sequences of V_L_ (accession number: U41645), and the constant region of the human kappa light chain (C_L_) were cloned into the pCAG-Ble vector. Antibody expression vectors were transfected into ExpiCHO-S cells using the ExpiCHO Expression System to produce humPcMab-47 and humPcMab-60. As a control human IgG_1_ (hIgG_1_) mAb, humCvMab-62 was generated from CvMab-62, an anti-SARS-CoV-2 spike protein S2 subunit mAb [22], using the same procedure. All antibodies were purified using Ab-Capcher (ProteNova Co., Ltd., Kagawa, Japan).

### 2.3. Flow Cytometry

Cells were washed with 0.1% BSA in PBS (blocking buffer) and treated with primary mAbs for 30 min at 4 °C, followed by treatment with anti-human IgG conjugated with fluorescein isothiocyanate (FITC) (Sigma-Aldrich Corp., St. Louis, MO, USA). Fluorescence data were collected using an SA3800 Cell Analyzer (Sony Corp., Tokyo, Japan).

### 2.4. Antibody-Dependent Cellular Cytotoxicity

The ADCC activity of humPcMab-60 was assessed as follows: Calcein AM-labeled target cells (CHO/PODXL, MIA PaCa-2, PK-45H, and Caco-2) were co-incubated with human natural killer (NK) cells (Takara Bio, Inc., Shiga, Japan) at an effector-to-target (E:T) ratio of 50:1 in the presence of 100 μg/mL of either control hIgG_1_ or humPcMab-60. Following a 4.5 h incubation period, the release of Calcein into the supernatant was quantified using a microplate reader (Power Scan HT; BioTek Instruments, Inc., Winooski, VT, USA).

Cytotoxicity was calculated as a percentage of lysis as described previously [22]. Data are presented as the mean ± standard error of the mean (SEM). Statistical significance was evaluated using a two-tailed unpaired *t*-test.

### 2.5. Complement-Dependent Cytotoxicity

The Calcein AM-labeled target cells (CHO/PODXL, MIA PaCa-2, PK-45H, and Caco-2) were plated and mixed with 100 μg/mL of control hIgG_1_ or humPcMab-60 and rabbit complement (final dilution 15%, Low-Tox-M Rabbit Complement; Cedarlane Laboratories, Hornby, ON, Canada). Calcein release into the medium was measured after incubation for 4.5 h at 37 °C.

### 2.6. Antitumor Activity of humPcMab-60

The Institutional Committee for Animal Experiments of the Institute of Microbial Chemistry approved the animal experiments (approval number: 2025-002). Humane endpoints for euthanasia were defined as body weight loss exceeding 25% of the original weight and/or a maximum tumor volume greater than 3000 mm^3^.

Female BALB/c nude mice were obtained from Jackson Laboratory Japan, Inc. Tumor cells (5 × 10^6^ cells) were subcutaneously injected into each mouse [26]. To evaluate the antitumor activity of humPcMab-60, 100 μg of humPcMab-60 or control hIgG_1_ was administered intraperitoneally to tumor-bearing mice on day 7 post-inoculation. A second dose was administered on day 14. In addition, human NK cells (5 × 10^5^ cells) were injected peritumorally on both days 7 and 14. Mice were euthanized on day 21 following tumor cell implantation.

Tumor size was measured as described previously [26]. Data are presented as the mean ± standard error of the mean (SEM). Statistical analysis was performed using one-way ANOVA followed by Sidak’s post hoc test. A *p*-value < 0.05 was considered statistically significant.

## 3. Results

### 3.1. Production of a Humanized Anti-PODXL CasMab, humPcMab-60

Our group previously generated a CasMab against PODXL, designated PcMab-60 (IgM, κ), by immunizing mice with soluble PODXL expressed in LN229 glioblastoma cells. PcMab-60 showed cancer-specific reactivity [24]. In contrast, a non-CasMab clone, PcMab-47, exhibited strong reactivity to both cancer and normal cells [19]. In this study, a humanized PcMab-60 (humPcMab-60) was engineered by fusing the V_H_ and V_L_ CDRs of PcMab-60 with the C_H_ and C_L_ chains of human IgG_1_, respectively (Figure 1A). The humPcMab-60 was mainly used in assays described in Figure 1A. PcMab-47 was also humanized and produced. As a control hIgG_1_ mAb, humCvMab-62 was produced from CvMab-62 (an anti-SARS-CoV-2 spike protein S2 subunit mAb). Under reduced conditions, the purity of the recombinant mAbs was confirmed by SDS-PAGE (Figure 1B).

As shown in Figure 2A, humPcMab-60 reacted with CHO/PODXL but not with parental CHO-K1. In contrast, PcMab-47 showed more potent reactivity to CHO/PODXL (Figure 2A). Similar reactivities were also observed in PDAC cell lines (MIA PaCa-2 and PK-45H) and the colorectal cancer cell line (Caco-2) (Figure 2B). Furthermore, PcMab-47 reacted with a lymphatic endothelial cell line, HDMVEC/TERT164-B. In contrast, humPcMab-60 did not (Figure 2C). These results indicate that humPcMab-60 retains cancer-specific reactivity.

### 3.2. ADCC, CDC, and Antitumor Effect by humPcMab-60 Against CHO/PODXL

The ADCC caused by humPcMab-60 against CHO/PODXL in the presence of human NK cells was investigated. As shown in Figure 3A, humPcMab-60 showed ADCC against CHO/PODXL (22.3% vs. 7.9% cytotoxicity of control hIgG_1_, *p* < 0.05). We then examined CDC caused by humPcMab-60 against CHO/PODXL in the presence of complements. As shown in Figure 3B, humPcMab-60 elicited CDC against CHO/PODXL (19.9% vs. 7.6% cytotoxicity of control hIgG_1_, *p* < 0.05). These results indicate that humPcMab-60 exerted ADCC and CDC against CHO/PODXL.

The antitumor activity of humPcMab-60 against CHO/PODXL xenografts was investigated. Following the inoculation of the CHO/PODXL, humPcMab-60 or control hIgG_1_ was intraperitoneally injected into CHO/PODXL xenograft tumor-bearing mice on days 7 and 14. Human NK cells were also injected around the tumors on days 7 and 14. The tumor volume was measured on days 7, 10, 14, 17, and 21 after inoculation. The administration of humPcMab-60 resulted in a significant reduction in CHO/PODXL xenografts on days 17 (*p* < 0.05) and 21 (*p* < 0.01) compared with that of control hIgG_1_ (Figure 3C). A significant reduction in xenograft weight caused by humPcMab-60 was observed in CHO/PODXL xenografts (57% reduction; *p* < 0.05; Figure 3D). Body weight loss was not observed in the xenograft-bearing mice (Figure 3E).

### 3.3. ADCC and CDC by humPcMab-60 Against Human Cancer Cells

The ADCC caused by humPcMab-60 against endogenous PODXL-expressing human cancer cell lines was further examined. Human NK cells were also used as effectors. As shown in Figure 4A, humPcMab-60 showed ADCC against MIA PaCa-2 (13.3% vs. 4.2% cytotoxicity of control hIgG_1_, *p* < 0.05), PK-45H (6.7% vs. 3.3% cytotoxicity of control hIgG_1_, *p* < 0.05), and Caco-2 (10.0% vs. 3.3% cytotoxicity of control hIgG_1_, *p* < 0.05). These results indicate that humPcMab-60 exerted ADCC against PODXL-expressing human cancer cell lines.

The CDC caused by humPcMab-60 against those cells in the presence of complements was examined. As shown in Figure 4B, humPcMab-60 elicited CDC against Caco-2 (14.4% vs. 4.1% cytotoxicity of control hIgG_1_, *p* < 0.05). However, a significant induction of CDC was not observed in MIA PaCa-2 and PK-45H. These results demonstrate that humPcMab-60 exerted significant CDC against Caco-2, but not other PDAC cell lines.

### 3.4. Antitumor Effect of humPcMab-60 Against Human Cancer Cells

Next, the antitumor activity of humPcMab-60 against MIA PaCa-2, PK-45H, and Caco-2 xenografts was investigated. Following the inoculation of those cancer cells, humPcMab-60 or control hIgG_1_ was intraperitoneally injected into the xenograft tumor-bearing mice on days 7 and 14. Human NK cells were also injected around the tumors on days 7 and 14. The tumor volume was measured on days 7, 10, 14, 17, and 21 after inoculation. The administration of humPcMab-60 resulted in a significant reduction in MIA PaCa-2 xenografts on days 14 (*p* < 0.05), 17 (*p* < 0.05), and 21 (*p* < 0.01) compared with control hIgG_1_ (Figure 5A). A significant reduction was also observed in the PK-45H xenograft on days 14 (*p* < 0.01), 17 (*p* < 0.01), and 21 (*p* < 0.01) (Figure 5B) and in the Caco-2 xenograft on days 10, 14 (*p* < 0.01), 17 (*p* < 0.01), and 21 (*p* < 0.01) (Figure 5C). A significant reduction in xenograft weight caused by humPcMab-60 was observed in MIA PaCa-2 (35% reduction; *p* < 0.01; Figure 5D), PK-45H (39% reduction; *p* < 0.01; Figure 5E), and Caco-2 (48% reduction; *p* < 0.01; Figure 5F). The MIA PaCa-2, PK-45H, and Caco-2 xenografts that were resected from mice on day 21 are also shown. Body weight loss was not observed in the xenograft-bearing mice (Figure 5G,H,I).

## 4. Discussion

PODXL has been a candidate of therapeutic target and a diagnostic biomarker in PDAC and colorectal cancers since the high PODXL expression is a potential indicator of poor prognosis [12,15]. PODXL could be detected in peripheral blood and used as a non-invasive diagnostic biomarker for the detection of PDAC [27]. In this study, it was demonstrated that a humanized anti-PODXL CasMab, humPcMab-60, could be a promising mAb-based tool in tumor therapy.

As of 2025, pancreatic cancer ranks as the fourth (for men) or third (for women) leading cause of estimated new cancer deaths in the United States [28]. PDAC is a common type of pancreatic cancer and is associated with an exceptionally poor prognosis with a 5-year survival rate of approximately 10% [29]. The most frequent oncogenic alterations—mutations in KRAS, CDKN2A, SMAD4, and TP53—are key drivers of PDAC pathogenesis [30,31]. Despite these common molecular events, PDAC represents a highly heterogeneous disease characterized by diverse histopathological features [32], molecular profiles [33], and clinical outcomes. Further investigation is required to determine the specific subtype(s) of pancreatic cancer in which PODXL is expressed.

Colorectal cancer ranks as the third (for men) or fourth (for women) leading cause of estimated new cancer deaths in the United States [28]. PODXL was identified as a target gene of β-catenin, which is frequently activated in colorectal cancer [34]. Furthermore, PODXL was upregulated by radiotherapy in both colorectal cancer tissues and cultured cells [35]. Radiation-induced PODXL promoted lamellipodia formation, migration, and invasiveness of colorectal cancer cells [35]. Therefore, humPcMab-60 could be used in combination with radiotherapy.

Although humPcMab-60 exhibited antitumor effects in vivo (Figure 3 and Figure 5), the reactivity of humPcMab-60 to CHO/PODXL, MIA PaCa-2, PK-45H, and Caco-2 was lower than that of a non-CasMab, humPcMab-47 (Figure 2). A similar phenomenon was observed in H_2_CasMab-2 (H_2_Mab-250), a CasMab against HER2. H_2_CasMab-2 differentially recognizes locally misfolded HER2 expressed on tumor cells compared with trastuzumab. Through the disruption of HER2 protein folding by dithiothreitol, HER2 recognition by H_2_CasMab-2 was significantly enhanced. In contrast, HER2 recognition by trastuzumab was significantly reduced [23]. A structure of H_2_CasMab-2 variable region complexed with an epitope peptide (amino acids 611–618) of HER2 was also determined. In the native state, this region of HER2 adopts an extended conformation, forming part of a β sheet [36,37]. Instead, when bound by H_2_CasMab-2, amino acids 611–618 undergo a bent conformation with little similarity to the native state [23]. The epitope of PcMab-60 was identified as a peptide sequence (_109-_RGGGSGNP_-116_) in PODXL, and the *K*_D_ value was determined as 4 × 10^−7^ M using surface plasmon resonance [25]. This result does not mean the low affinity among mAbs examined in our laboratory. Therefore, we propose the idea that the epitope is partially exposed in cancer cells but not in normal cells. Further studies are required to investigate whether humPcMab-60 recognizes the misfolded structure of PODXL and clarify the structure of the humPcMab-60-PODXL complex.

Since PcMab-60 did not cross-react with mouse PODXL (Appendix A), it is difficult to predict the effect of humPcMab-60 to mouse normal tissues. PODXL is expressed in normal cells such as vascular and lymphatic endothelial cells [5]; the evaluation of the in vivo toxicity of humPcMab-60 is essential for further development. The investigation of the toxicity of humPcMab-60 against cynomolgus monkeys is essential for the prediction of side effects.

Aberrant glycosylation is a hallmark of malignancies and contributes to the generation of tumor-specific glycosylated epitopes [38]. PODXL-targeting mAbs that selectively recognize cancer-associated glycosylated epitopes, while sparing PODXL expressed on normal tissues, have been developed [39]. Among these, PODO447 demonstrates remarkable specificity for a tumor-associated glycosylated epitope of PODXL and does not cross-react with normal adult human tissues. Epitope mapping using glycosylation-deficient cell lines identified the recognized epitope as an *O*-linked core 1 glycan presented in the structural context of the PODXL polypeptide backbone [39]. The PcMab-60 epitope (_109-_RGGGSGNP_-116_) can be modified by *N*- and/or *O*-glycosylation. PcMab-60 can recognize a non-glycosylated synthesized peptide in an enzyme-linked immunosorbent assay and surface plasmon resonance analysis [25]. Further analyses of glycosylation at these sites are needed. Moreover, the difference in glycosylation between cancer and normal cells and/or between cultured cells and the xenograft should be investigated to clarify the mechanism of cancer-specific recognition by PcMab-60.

Circulating tumor cells (CTCs), recognized as initiators of metastasis, are present in the bloodstream either as individual cells or as multicellular clusters, the latter of which demonstrate significantly higher metastatic potential than single CTCs [40]. PODXL is known to facilitate CTC cluster formation [41]. PODXL expression was found to be elevated in CTC clusters compared to single CTCs isolated from blood samples of breast cancer patients [42]. Furthermore, genetic silencing of PODXL or treatment with an anti-PODXL mAb markedly suppressed tumor cell clustering in vivo and effectively inhibited metastatic colonization following intravenous injection into mice [41]. It is worthwhile to investigate whether humPcMab-60 can recognize PODXL on the CTC cluster and reduce metastatic colonization in the mouse model.

Notably, the loss of terminal sialylation in glycoproteins within CTC clusters contributes to cellular dormancy, facilitates resistance to chemotherapy, and enhances metastatic potential [41]. PODXL knockdown reversed the tumor cell aggregation induced by the knockout of β-galactoside α2,6-sialyltransferase 1 (ST6GAL1) [41], which catalyzes the addition of α2,6-sialic acid onto terminal glycans on glycoproteins [43]. These results suggest that PODXL is a potential target for counteracting the metastasis of quiescent tumor cells. Our group developed more than one hundred anti-PODXL hybridoma clones (PcMabs), and a part of PcMabs has been updated at Antibody Bank (http://www.med-tohoku-antibody.com/topics/001_paper_antibody_PDIS.htm, accessed on 6 August 2025). Our PcMabs, including PcMab-60, may contribute to the identification of quiescent PODXL-positive tumor cells and the development of therapeutic applications to target those cells.

## Figures and Tables

**Figure 1 antibodies-14-00067-f001:**
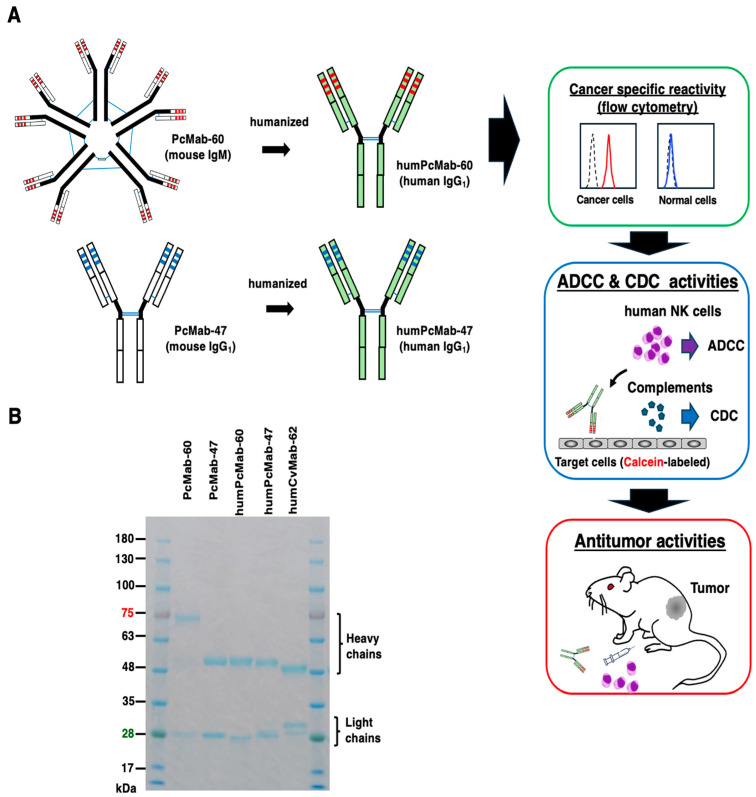
Production of humPcMab-60. (**A**) Human IgG_1_ mAbs, humPcMab-60 and humPcMab-47, were generated from PcMab-60 (mouse IgM) and PcMab-47 (mouse IgG_1_), respectively. Using humPcMab-60, the cancer-specific reactivity, ADCC, CDC, and antitumor effect were investigated. (**B**) PcMab-60, PcMab-47, humPcMab-60, humPcMab-47, and humCvMab-62 were treated with sodium dodecyl sulfate sample buffer containing 2-mercaptoethanol. Proteins were separated on a polyacrylamide gel. The gel was stained with Bio-Safe CBB G-250 Stain.

**Figure 2 antibodies-14-00067-f002:**
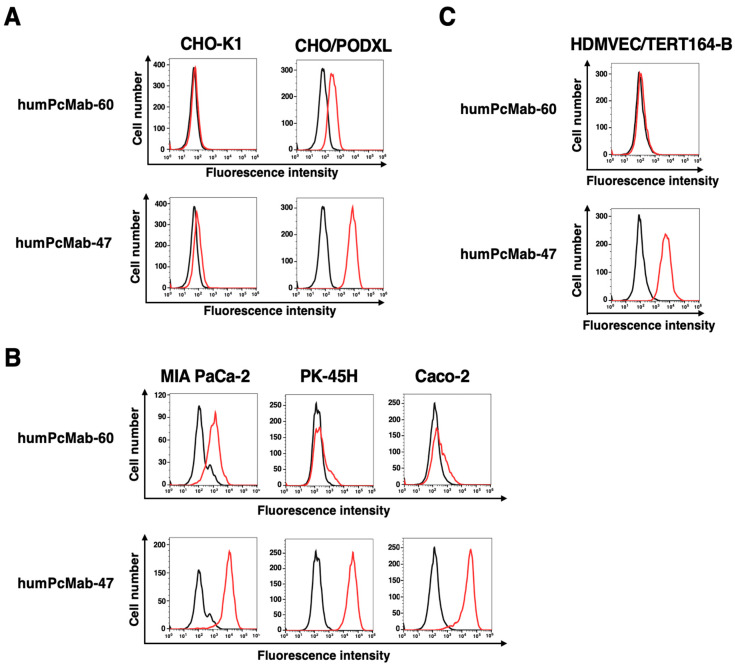
Flow cytometry analysis of humPcMab-60 and humPcMab-47 to tumor and normal cells. CHO-K1 and CHO/PODXL (**A**), PDAC cell lines (MIA PaCa-2 and PK-45H) and the colorectal cancer cell line (Caco-2) (**B**), and a lymphatic endothelial cell line (HDMVEC/TERT164-B) (**C**) were treated with 10 µg/mL of humPcMab-60 or humPcMab-47. Then, the cells were treated with FITC-conjugated anti-human IgG. Data were collected and analyzed using the SA3800 Cell Analyzer.

**Figure 3 antibodies-14-00067-f003:**
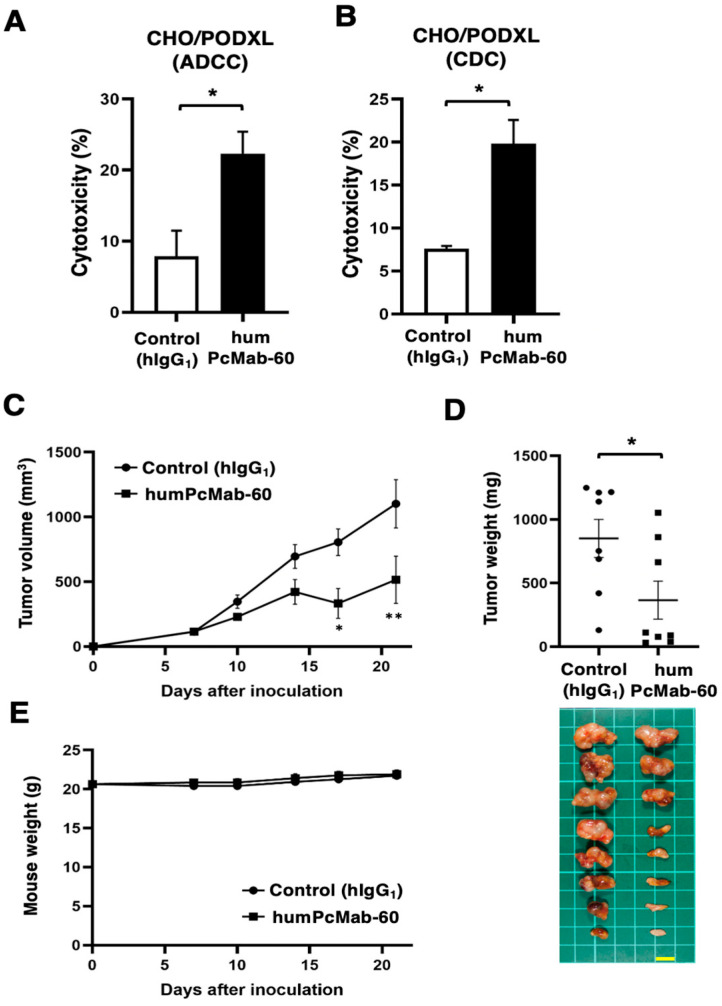
ADCC, CDC, and an antitumor effect by humPcMab-60 against CHO/PODXL xenografts. (**A**) ADCC induced by humPcMab-60 or control human IgG_1_ (hIgG_1_) against CHO/PODXL. (**B**) CDC induced by humPcMab-60 or control hIgG_1_ against CHO/PODXL. Values are shown as the mean ± SEM. Asterisks indicate statistical significance (* *p* < 0.05; two-tailed unpaired *t* test). (**C**) Antitumor activity of humPcMab-60 against CHO/PODXL xenografts. CHO/PODXL was subcutaneously injected into BALB/c nude mice (day 0). An amount of 100 μg of humPcMab-60 or control hIgG_1_ was intraperitoneally injected into each mouse on days 7 and 14. Human NK cells were also injected around the tumors. The tumor volume is presented as the mean ± SEM. ** *p* < 0.01; * *p* < 0.05 (ANOVA with Sidak’s multiple comparisons test). (**D**) The tumor weights were measured on day 21. Values are presented as the mean ± SEM. * *p* < 0.05 (two-tailed unpaired *t* test). (**E**) Body weights of the xenograft-bearing mice treated with humPcMab-60 or control hIgG_1_. There is no statistical difference.

**Figure 4 antibodies-14-00067-f004:**
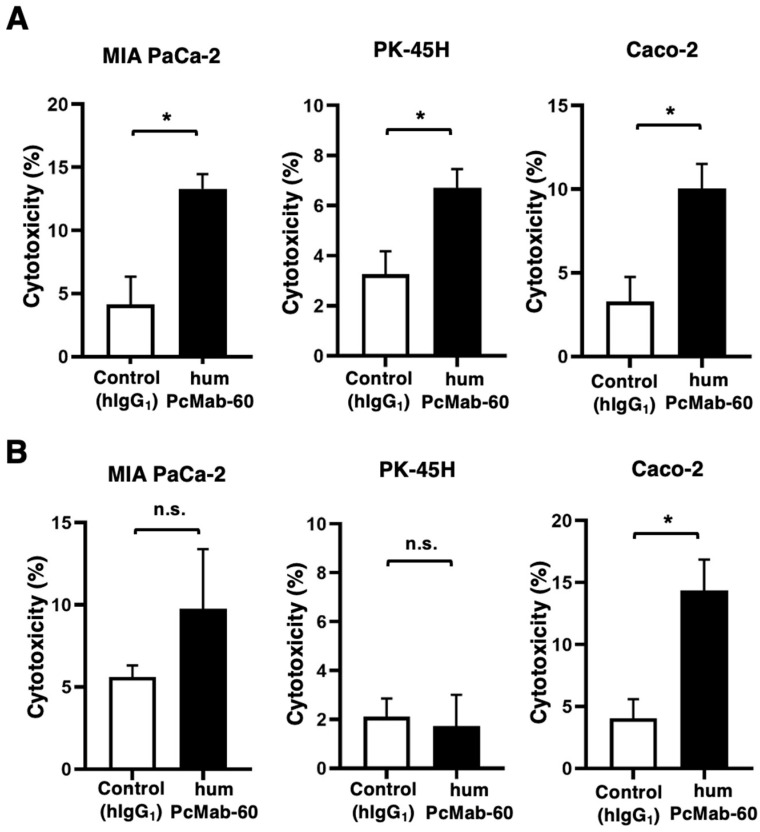
ADCC and CDC by humPcMab-60 against human cancer cell lines. (**A**) ADCC against MIA PaCa-2, PK-45H, and Caco-2 in the presence of humPcMab-60 or control hIgG_1_. (**B**) CDC against MIA PaCa-2, PK-45H, and Caco-2 in the presence of humPcMab-60 or control hIgG_1_. Values are shown as the mean ± SEM. Asterisks indicate statistical significance (* *p* < 0.05; two-tailed unpaired *t* test). n.s., not significant.

**Figure 5 antibodies-14-00067-f005:**
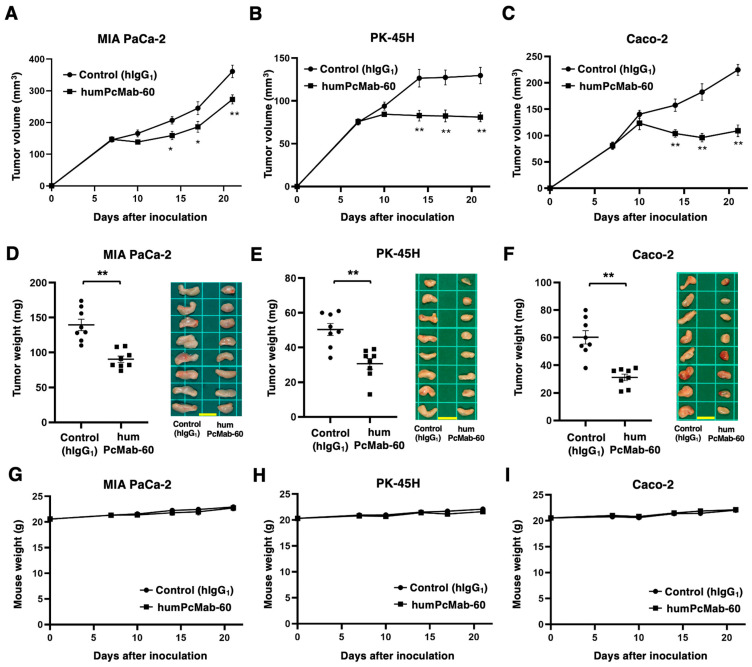
Antitumor activity of humPcMab-60 against human cancer xenografts. (**A**–**C**) MIA PaCa-2 (**A**), PK-45H (**B**), and Caco-2 (**C**) were subcutaneously injected into BALB/c nude mice (day 0). An amount of 100 μg of humPcMab-60 or control hIgG_1_ was intraperitoneally injected into each mouse on days 7 and 14. Human NK cells were also injected around the tumors. The tumor volume is presented as the mean ± SEM. ** *p* < 0.01; * *p* < 0.05 (ANOVA with Sidak’s multiple comparisons test). (**D**–**F**) The tumor weights of MIA PaCa-2 (**D**), PK-45H (**E**), and Caco-2 (**F**) xenografts were measured on day 21. Values are presented as the mean ± SEM. ** *p* < 0.01 (two-tailed unpaired *t* test). The resected MIA PaCa-2, PK-45H, and Caco-2 xenograft tumors are shown (scale bar, 1 cm). (**G**–**I**) Body weights of MIA PaCa-2 (**G**), PK-45H (**H**), and Caco-2 (**I**) xenograft-bearing mice treated with humPcMab-60 or control hIgG_1_. There is no statistical difference.

## Data Availability

The data presented in this study are available in the article and Appendix A.

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
