# Peer review of "A Cancer-Specific Anti-Podocalyxin Monoclonal Antibody (humPcMab-60) Demonstrated Antitumor Efficacy in Pancreatic and Colorectal Cancer Xenograft Models"

_2073-4468, 2025, doi:10.3390/antib14030067_

Round 1

Reviewer 1 Report

Comments and Suggestions for Authors

The study "A cancer-specific anti-podocalyxin monoclonal antibody (humPcMab-60) demonstrated antitumor efficacy against human cancer xenografts" claims that humanized  anti-PODXL mAb, PcMab-60 was cancer cell specific, did not bind to endothelial cells and worked effectively in mouse xenograft model. The concept of CasMab is interesting and cancer cell specificity of PcMab-60 is very attractive and can open a whole new horizon. However, important details of the antibody are missing and several points need to be clarified to make the paper of value to readers in the field:

(1) Humanization of PcMab-60 is mentioned without any details about the donor or acceptor germlines. Lines 186-188 mentions "In this study, we engineered a humanized PcMab-60 (humPcMab-60) by fusing the VH and VL CDRs of PcMab60 with the CH and CL chains of human IgG1, respectively (Figure 1A)." It is not clear how and which human frameworks were selected and reasons for the same. 

(2) PcMab-60 is IgM derived and expected to be of low affinity. There is no discussion in this paper of the role of this low affinity in what is being claimed as cancer specificity. There should be a detailed disclosure of podocalyxin expression level in cancer cells vs endothelial cells. There is a mention of this in this manuscript, nor in ref 27 or other references therein. Therefore, to claim that PcMab-60 is cancer specific one has to demonstrate precise expression levels on cancer and normal endothelial cells. If expression levels are similar, there should be an unambiguous investigation to unravel why PcMab-60 binds to the cancer and not the normal endothelial cells. This is not at all clear in this paper and is a serious void. 

(3) It is also not clear from ref 20 (which is not freely accessible) how the linear epitope of PcMab-60 differs between normal and cancer cells. If there is no difference, then for this paper to be of value, as mentioned above, accurate expression levels of the protein and its splice or post translational forms on normal and cancer cells is critically important to understand why humPcMab-60 might be specific to cancer cell and not bind to normal cells.

(4) There is no effort at studying the cross reactivity to mouse podocalyxin of PcMab-60 or its "humanized" variant used in the study. This is another important to support the claim that humPcMab-60 reacts only to cancer cells. If the antibody cross reacts with mouse podocalyxin and its expression pattern is similar to humans, then the results of this study can take a whole positive turn. But without that data, the study remains ambiguous.

Reviewer 2 Report

Comments and Suggestions for Authors

The authors aim to generate a humanized immunoglobulin (IG) with antitumor activity in mouse models and various human and mouse cell lines for colorectal cancer and pancreatic ductal adenocarcinoma.

This is a well-founded study, but it requires modifications that would improve reading and comprehension:

  1. The title should be more explanatory, indicating the types of cancer, the types of cancer analyzed, and the types of cancer studied.
  2. The authors should write the manuscript in an impersonal style.
  3. They should discuss the limitations of the study.
  4. There is some confusion in the introduction, methodology, and results when using mouse or human biological material; this is an important aspect. Perhaps an outline at the beginning of the manuscript could clarify or provide an overview of the experiments performed.
  5. Numerous statements made in the introduction lack references. Furthermore, the introduction should focus on the study performed.
  6. Line 120? What does 95% air mean? This aspect is not usually indicated in this way.
  7. Abbreviations should be indicated first in the text and avoided in titles and subtitles.
  8. The manuscript would benefit from a timeline of the experiments.
  9. Avoid repeating the results obtained and citing figures in the discussion.
  10. Authors should indicate the controls used in their study in the methodology.

Reviewer 3 Report

Comments and Suggestions for Authors

The present study investigates the efficacy of humPcMab-60, which was developed expressly for the targeting of human cancer cells and tumours.

The study is developing a version of a PODXL-targeted antibody (CasMab - humPcMab-60) that recognises only cancer cells, a highly expressed antibody in cancer cells but also found in normal tissues. This represents a substantial advancement in the field of preventing on-target off-tumor toxicities.

The production, purification and testing procedures of the recombinant antibody are described in detail, which increases the reproducibility of the study.

The efficacy of the antibody was evaluated in terms of both ADCC (antibody-dependent cellular cytotoxicity) and CDC (complement-dependent cytotoxicity). In addition, its effect on tumour growth in mice was demonstrated.

The manuscript has been written meticulously and carefully, the results are well summarised, and possible reasons for the study's limitations (namely, that the CDC was only significant in the Caco-2 cell line and was ineffective in pancreatic cancer cells; that the humPcMab-60 antibody did not show as strong reactivity as humPcMab-47) are given in the discussion section.

In vivo studies were conducted in accordance with the ARRIVE guidelines.

Author Response

Thank you very much.

Round 2

Reviewer 1 Report

Comments and Suggestions for Authors

Thank you for trying to address the questions. Appreciate disclosing the limitation of the study around lack of cross reactivity with mouse tissue and what needs to be done with pre-clinical models before further perusal. It is important to highlight specially given the sensorgrams in Fig2B that humPcMab-60 it is likely to be of lower affinity compared to humPcMab-47 and may be the reason for the differential binding to human cancer cells. The claim that humPcMab-60 is cancer specific is purely presumptive at this stage. This is a critical gap in the study.

Lastly, since the paper talks about humanization, it is only fair to describe it in detail, including which germline frameworks were used and why and if there has been any back mutation needed, along with any change in binding properties between the wt and humanized version. 

Author Response

Thank you for trying to address the questions. Appreciate disclosing the limitation of the study around lack of cross reactivity with mouse tissue and what needs to be done with pre-clinical models before further perusal. It is important to highlight specially given the sensorgrams in Fig2B that humPcMab-60 it is likely to be of lower affinity compared to humPcMab-47 and may be the reason for the differential binding to human cancer cells. The claim that humPcMab-60 is cancer specific is purely presumptive at this stage. This is a critical gap in the study.

We previously assessed the affinity of PcMab-60 to the epitope peptide using SPR (Monoclon Antib Immunodiagn Immunother 2021;40: 227-232). According to the results, the KD value was 4 x 10-7 M, which does not mean the low affinity among mAbs examined in our laboratory. Therefore, we discussed that the cancer specific epitope could be partially exposed in cancer cells, but not in normal cells.
We revised the discussion as follows.

Page 9, Line 309
The epitope of PcMab-60 was identified as a peptide sequence (109-RGGGSGNP-116) in PODXL, and the KD value was determined as 4 × 10-7 M using surface plasmon resonance [25]. It does not mean the low affinity among mAbs examined in our laboratory. Therefore, we propose a possibility that the epitope is partially exposed in cancer cells, but not in normal cells.

Lastly, since the paper talks about humanization, it is only fair to describe it in detail, including which germline frameworks were used and why and if there has been any back mutation needed, along with any change in binding properties between the wt and humanized version.

We added the accession number of human IgG framework in section 2.2.